# Mechanical Mastication Reduces Fuel Structure and Modelled Fire Behaviour in Australian Shrub Encroached Ecosystems

**Madeleine A. Grant** [1,*], **Thomas J. Duff** [1,2], **Trent D. Penman** [1], **Bianca J. Pickering** [1] **and Jane G. Cawson** [1]

1   School of Ecosystem and Forest Sciences, Faculty of Science, The University of Melbourne, Baldwin Spencer Building, Parkville, VIC 3010, Australia; thomas.duff@cfa.vic.gov.au (T.J.D.); trent.penman@unimelb.edu.au (T.D.P.); bee.pickering@unimelb.edu.au (B.J.P.); jane.cawson@unimelb.edu.au (J.G.C.)
2   Bushfire Management, Country Fire Authority, Burwood, VIC 3125, Australia
*   Correspondence: madeleine.a.grant@gmail.com

**Abstract:** Shrub encroachment of grassland and woodland ecosystems can alter wildfire behaviour and threaten ecological values. Australian fire managers are using mechanical mastication to reduce the fire risk in encroached ecosystems but are yet to evaluate its effectiveness or ecological impact. We asked: (1) How does fuel load and structure change following mastication?; (2) Is mastication likely to affect wildfire rates of spread and flame heights?; and (3) What is the impact of mastication on flora species richness and diversity? At thirteen paired sites (masticated versus control; $n = 26$), located in Victoria, Australia, we measured fuel properties (structure, load and hazard) and floristic diversity (richness and Shannon's $H$) in 400 mP$^2$ plots. To quantify the effects of mastication, data were analysed using parametric and non-parametric paired sample techniques. Masticated sites were grouped into two categories, 0–2 and 3–4 years post treatment. Fire behaviour was predicted using the Dry Eucalypt Forest Fire Model. Mastication with follow-up herbicide reduced the density of taller shrubs, greater than 50 cm in height, for at least 4 years. The most recently masticated sites (0–2 years) had an almost 3-fold increase in dead fine fuel loads and an 11-fold increase in dead coarse fuel loads on the forest floor compared with the controls. Higher dead coarse fuel loads were still evident after 3–4 years. Changes to fuel properties produced a reduction in predicted flame heights from 22 m to 5–6 m under severe fire weather conditions, but no change in the predicted fire rate of spread. Reductions in flame height would be beneficial for wildfire suppression and could reduce the damage to property from wildfires. Mastication did not have a meaningful effect on native species diversity, but promoted the abundance of some exotic species.

**Keywords:** fire management; mechanical fuel treatment; mulching; fuel hazard; woody weeds; invasive native shrub; wildfire





## 1. Introduction

Shrub encroachment, where grassland or woodland ecosystems become heavily dominated by one or two shrub species, is a growing concern for ecosystem managers [1,2]. This vegetation shift may be caused by a number of factors including altered fire regimes [3–5], changes to grazing pressures [5–7], farm land abandonment [8], the extirpation of apex predators and key herbivore populations [9,10] and a changing climate [11,12].

Shrub encroachment is generally thought to disrupt ecosystem processes and services [1]. In eucalypt woodlands and forests, it has been observed to reduce the ecological value of a site by decreasing the flora species richness [13], and threaten ecologically sensitive remnant patches of vegetation [5]. However, all ecological values are not compromised by this shift in plant composition [1,14] as dense shrubs can provide an important habitat for fauna species [15,16].

For fire managers, shrub encroachment is a problem as the dense shrub poses a heightened wildfire risk [3,17,18]. The challenge is two-fold. Firstly, the encroaching

shrubs can intensify fire behaviour by changing the fuel structure. The shrubs raise the elevated fuel load, which causes increased flame heights and makes fire suppression more difficult [19–21]. This is particularly concerning when the shrub encroachment occurs close to urban settlements where fire has the potential to damage or destroy houses. Secondly, the dense shrub layer limits the ability of fire managers to use conventional fuel reduction techniques like prescribed burning. This is because fire will only propagate through dense shrubby fuels under windy conditions [17,22] when it is difficult to safely implement a prescribed burn.

Fire managers are compelled to consider alternative methods for fuel hazard reduction to reduce shrub density in shrub encroached woodlands and forests. One such method is mechanical mastication, which uses heavy machinery to mulch or chip the understory trees and shrubs and redistribute them as debris on the forest floor [23–25]. Mastication is widely used in North American coniferous forests and shrublands [26] where it has been found to reduce flame heights and fire intensities [26,27], which can increase suppression opportunities and improve the resilience of forest stands after wildfire [28,29].

In temperate Australia, shrub encroachment is a growing issue for land and fire managers. Spatial analyses indicate the area impacted by shrub encroachment is expanding beyond localised infestations [6,7,30] to landscape scale proportions [31]. For example, a 2% increase in woody vegetation cover in woodland ecosystems was estimated to have occurred in the State of Victoria over 16 years (1989–2007). Fire managers have started using mechanical mastication over the last few years to manage the fire risk associated with shrub encroachment.

Yet, there is an absence of research on the efficacy of mechanical mastication as a fuel management technique in Australian vegetation [32]. Mechanical 'thinning' in Australia has been found to reduce fine fuels overtime [33] and lessen fire-line intensity and rate of spread in wildfire simulations [34]. However, these thinning techniques differ from mechanical mastication as the thinned material is not retained as a mulch layer on the forest floor. Anecdotally, mastication is effective in reducing flame heights and assisting with fire suppression. For example, during a wildfire in the The Pines Flora and Fauna Reserve (in the urban-wildland interface on the Mornington Peninsula in Victoria), fire crews reported lower flame heights in masticated areas, though rates of spread were not reduced. Direct attack was not possible in the masticated or untreated shrubby areas, but crews were able to defend houses more easily adjacent to masticated fuel because lower flame heights made asset protection safer to implement [35].

Studies about the effectiveness of fuel treatments are crucial for fire managers to make evidence-based decisions about resource allocation to reduce wildfire risk and manage ecosystems. As part of this decision-making process, fire managers need an understanding of both the reductions in wildfire behaviour and any detrimental environmental outcomes that are likely to occur. We sought to quantify the effectiveness of mechanical mastication as a fuel management technique in shrub encroached, south-eastern Australian woodlands and open forests. Follow up herbicide was considered a component of the mastication treatment where the encroaching shrub species were re-establishing. In this study, we asked:

- How does fuel load and structure change with mastication in shrub encroached woodlands and forests?
- Is mastication likely to affect wildfire rates of spread and flame heights?
- What is the impact of mastication on flora species richness and diversity?

We hypothesise that mastication will have a substantial impact on fuel structure, relocating the shrub layer to the ground, but not on overall fuel loads. We expect that these changes to fuel structure will translate to reductions in modelled flame heights and rates of spread. Ecologically, we expect to see an increase in flora species richness and diversity as a result of mastication, as the removal of the dominant shrub layer provides an opportunity for a broader diversity of species to establish.

## 2. Materials and Methods

### 2.1. Study Area

The study evaluated sites that had been mechanically masticated for fuel management purposes. We contacted land managers in Victoria, Australia and identified 13 sites that were recently treated (in the last 4 years) and had been affected by shrub encroachment prior to mastication. Victoria has a temperate climate, with warm to hot and dry summers [36]. Vegetation at the sites was classified as either eucalypt woodland, open eucalypt forest or shrubland (Table 1). Woodland is defined by having a sparse canopy cover (10–30%) of eucalypt trees, whereas open forest has a moderately dense canopy (30–70%) [37]. Shrubland has no tree canopy. Each of the sites shared a common pattern of shifting understorey vegetation composition by one or more of the following shrub species: Coast Teatree (*Leptospermum laevigatum*), Coast Wattle (*Acacia longifolia* subsp. *sophorae*) and Sallow Wattle (*Acacia longifilia* subsp. *longifolia*). Each shrub species is capable of growing to tree-like dimensions, 5 m in height, increasing to 10 m for the *Acacia* spp. [38] Regeneration is primarily via seed, germinating prolifically post disturbance (mechanical or fire) [39–41]. For the *Leptospermum* sp., this occurs via the simultaneous release of unshed seed from its woody fruits and for the *Acacia* spp. the mass germination of soil-stored seed. Germinable seeds persist for up to one year and over 10 years, respectively. All three species are capable of re-sprouting after mechanical disturbance.

### 2.2. Fuel Treatment

Mastication was completed between 2014 and 2018, using either a 'mulching' or 'slashing' attachment on a posi-track, skid steer or walking excavator (Table 1). We sought to reduce any variability that resulted from the mastication technique by sampling sites where the resultant debris was evenly distributed, as opposed to being windrowed. Targeted follow-up treatments had been applied at five out of the 13 sites, where the invasive shrubs were re-establishing. This involved spot spraying with a broadleaf herbicide, cut and paint with glyphosate or manual removal (Table 1). Follow-up treatments were not applied to all sites because in some instances the local land manager did not consider the amount of regrowth substantial enough to warrant the cost. Since mastication is a relatively new technique in Victoria, there were not enough sites available to independently evaluate the effect of herbicide treatment, mastication technique or treatment season. As we were unable to quantify the individual effects of mastication-only versus mastication plus herbicide, we considered the follow up herbicide a part of the mastication treatment. Data were grouped into two categories to evaluate the effects of time since mastication: 0–2 years post mastication and 3–4 years post mastication.

### 2.3. Site Selection

We used a paired study design, masticated versus control, at each site. The control plots were shrub encroached and selected to match the fuel condition in the masticated plots prior to treatment. This was determined by consulting local land managers and using aerial imagery. Paired plots were located within 500 m of each other to minimise environmental variation between plots. Our sample size and number of plots within each 'time since mastication' category was limited by the availability of masticated and control areas.

A random compass bearing and distance was generated from the nearest road to determine the north-west corner of each 400 m$^2$ plot (20 × 20 m) within each predefined treatment area. A minimum 10 m buffer was established between the plot and any roads to reduce edge effects. Masticated areas are often small and therefore 10 m was frequently the maximum buffer width that could fit within the site. Figure 1 illustrates the typical difference in appearance between the paired masticated and control plots, 0–2 years post mastication.

**Table 1.** Details about the masticated plots.

| Site | UTM Grid Zones | Region | Year | Area | Plant | Follow Up | Shrub Species | Veg Class [B] |
|---|---|---|---|---|---|---|---|---|
| West-Soup Trk | H 55 E 0436507 N 5693030 | Wilsons Prom | 2018 | 0.5 ha | 110 Terex Positrack | N/A | Coast Teatree | Coastal shrubland |
| Pohlners Rd | H 54 E 0624050 N 5913495 | Grampians | 2018 | Unkn | 110 Terex Positrack | N/A | Sallow Wattle | Open forest |
| Tamarisk Dr | H 55 E 0339318 N 5779151 | Mornington Peninsula | 2018 | 2 ha | Skid Steer | N/A | Coast Teatree | Woodland |
| Tecoma Rd | H 54 E 0552896 N 5751114 | Far South West | 2017 | 0.5 ha | Unspec. | N/A | Coast Wattle | Woodland |
| Copper-mine Trk | H 54 E 0626409 N 5912772 | Grampians | 2017 | Unkn | 110 Terex Positrack | N/A | Sallow Wattle | Woodland |
| Pipeline Trk | H 55 E 0254627 N 5747230 | Otway Plain | 2017 | 8 ha | 100 HP Skid Steer[S] | N/A | Coast Wattle | Woodland |
| PMR Cave Rd | H 54 E 0499452 N 5795604 | Far South West | 2016 | 1.5 ha | Unspec. | N/A | Coast Wattle | Woodland |
| Mt Zero Rd | H 54 E 0623371 N 5917246 | Grampians | 2015 | Unkn | 110 Terex Positrack | Glyphosate Cut/Paint | Sallow Wattle | Woodland |
| Roses Gap Rd | H 54 E 0630116 N 5908325 | Grampians | 2015 | Unkn | 110 Terex Positrack | N/A | Sallow Wattle | Woodland |
| Darnley Trk | H 55 E 0340993 N 5779495 | Mornington Peninsula | 2015 | 7 ha | Skid Steer | Broad leaf Spot Spray | Coast Teatree | Woodland |

**Table 1.** *Cont.*

| Site | UTM Grid Zones | Region | Year | Area | Plant | Follow Up | Shrub Species | Veg Class [B] |
|------|---------------|--------|------|------|-------|-----------|---------------|---------------|
| Odonahue Rd | H 55 N 0252422 E 5742949 | Otway Plain | 2014 | 2 ha | 100 HP Skid Steer [S] | Glyphosate Cut/Paint | Coast Wattle | Woodland |
| Arthurs Seat | H 55 E 0320957 N 5753328 | Mornington Peninsula | 2014 | 1 ha | Walking excavator | Broad leaf Spot spray | Coast Teatree | Open forest |
| Waterfall Gully Rd | H 55 E 0319490 N 5750218 | Mornington Peninsula | 2014 | 7 ha | Walking excavator | Broad leaf Spot spray | Coast Teatree | Open forest |

N/A—herbicide treatment not required. [S] 'Slasher' attachment used. [B] Vegetation classes based on the National Vegetation Information System [42].

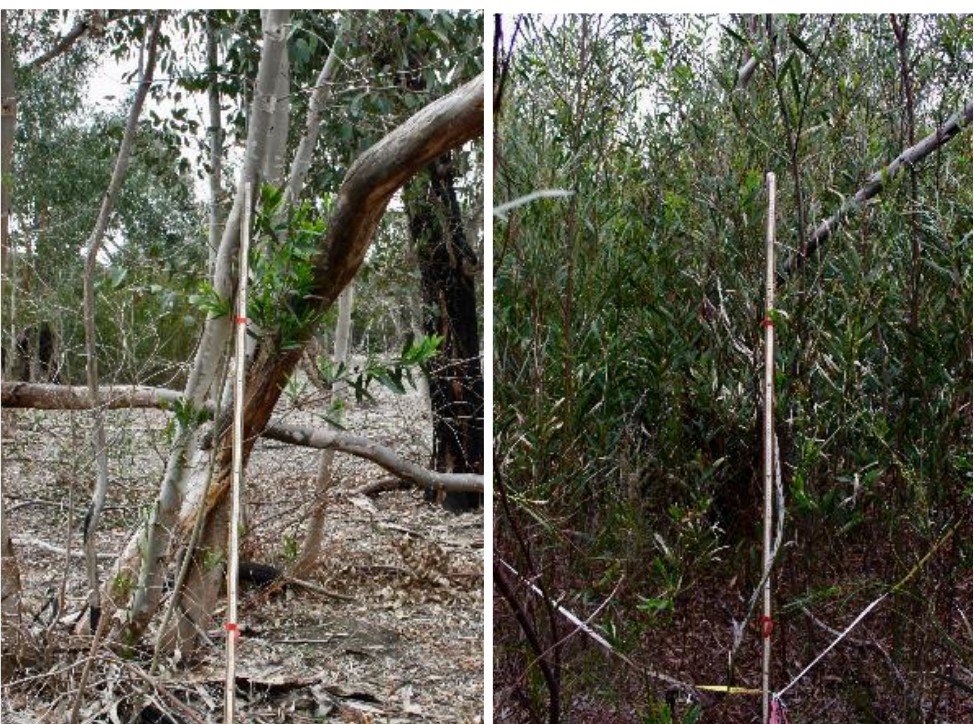

**Figure 1.** Paired plots with masticated (**left**) and control (**right**). Pohlners Rd site at the Grampians National Park masticated in 2018. Images feature a 2-m height pole. Photos taken in November 2018.

*2.4. Field Measurements*

All measurements took place in 2018 from November to early December. Shrub density was measured using the point-intercept method [43] on a 5 m grid within the plot. We recorded the presence/absence of combined live fine fuel (<2 mm thick) and dead fine fuel (<6 mm thick) within each height range (0–10, 10–20, 20–30, 30–40, 40–50, 50–100, 100–150, 150–200 cm) using a 2 m height pole. The presence of fine fuel at each height range was summarised as a proportion of the total 25 measurements taken within each plot. We measured canopy closure (combined understorey and tree canopy) at each corner, facing towards the centre of the plot, using a spherical densiometer (Model A). The four readings were averaged for each plot, hereafter referred to as vegetation closure. Cover abundance (%) was visually estimated for each flora species present within the plot.

The fuel load of the litter bed was measured by collecting all dead fuel particles < 25 mm in diameter (leaves, twigs, bark and woody fragments) inside a 0.1 m$^2$ sampling ring at each of the four corners of the plot [44]. Fuel was separated into dead fine (<6 mm diameter) and dead coarse (6 ≥ diameter) size classes. Fuel samples were dried in an oven set to 105 °C until a constant weight was achieved (>24 h) and then weighed to determine the mass per unit area.

Fuel hazard ratings were measured for each plot using the Overall Fuel Hazard (OFH) Assessment Guide [19], which is the standard approach used in the study area [45]. Four fuel strata (bark, elevated, near-surface and surface) are assessed separately and assigned a categorical fuel hazard rating (low, moderate, high, very high or extreme). The rating is based on a visual assessment of cover and continuity (horizontal and vertical) of dead and live fine fuel and some simple measurements such as surface fuel depth, which was measured 12 times within the plot, and the average height for elevated and near-surface fuels were also recorded as part of this assessment. An aggregate rating (overall fuel hazard) is produced by combining the visual assessments of the four fuel strata. This process was aided by the descriptions and images provided in the OFH Guidebook [19] and by maintaining the same person for all assessments to remove observer bias [46].

### 2.5. Data Analysis

Nine metrics were calculated from the field data to represent fuel properties and floristic diversity (Table 2). The metrics for the masticated treatments and control plots were visually summarised using boxplots, which also depict the mean. We used paired tests to evaluate the statistical significance of differences between the masticated and control plots. Paired *t*-tests were used where the data were normally distributed while Wilcoxon Signed-Rank tests were used when the data were not normally distributed. Data were tested for normality using the Shapiro–Wilk test [47].

We used generalised additive models (GAMs) for the shrub density data, to fit a curve to show the trend in shrub vertical density for masticated versus control plots [48]. This modelling approach fits a smooth spline to allow the visualisation of trends in the data and prevents the need to make prior assumptions about the form of any trends. The GAMs were fit as cubic splines and we allowed a maximum of four degrees of freedom. In the fitting process, a penalty term was applied, which limited the complexity of the fit based on restricted maximum likelihood (REML) criteria.

We summarised the categorical fuel hazard data using contingency tables and tested for a statistically significant association between the hazard classes and the treatments using Fisher's exact test (as the sample size was too small for a Chi-square test). All data analyses were conducted using the statistical software package R, version 3.5.1 [49], with library packages; doBy [50]; mgcv [48]; dplyr [51]; car [52].

**Table 2.** Variables derived from field data used to quantify fuel properties and floristic diversity in masticated and control plots.

| Variables | Units | Description |
|---|---|---|
| Shrub density | % | Percent of vertical fuel present at 8 height increments |
| Vegetation closure | % | Mean vegetation closure taken from a measurement at each corner per plot |
| Fuel hazard rating | – | Rating from low to extreme for each fuel strata (bark, elevated, near-surface, surface) and overall fuel hazard rating as per the OFH Assessment Guide [19] |
| Surface fuel depth | mm | Mean fuel depth taken from 12 measurements per plot |
| Surface fuel load | t ha$^{-1}$ | Mean surface fuel load taken from four measurements per plot, two size classes: Dead fine (<6 mm diameter)/Dead coarse (6 ≥ diameter < 25 mm) |
| Species richness | – | Number of species recorded per plot |
| Shannon's diversity($H'$) | – | Index of species diversity derived from species cover abundance recorded per plot. It reflects the relative contribution of each species, where an even distribution of abundance among species receives a higher value of diversity [53,54]. |
| Species richness—excl. exotics | - | Number of species recorded per plot, excluding non-native species |
| Shannon's diversity ($H'$)—excl. exotics | - | Index of species diversity derived from species cover recorded per plot, excluding non-native species |

### 2.6. Fire Modelling

Fire behaviour modelling was used to estimate how changes in fuel structure caused by mastication may affect fire behaviour. We used the fuel hazard rating version of the Dry Eucalypt Forest Fire Model (also known as Vesta) [20], which has been recommended for operational use across Australia [55,56]. Measured surface and near-surface fuel hazard rating and elevated fuel height were included as inputs to the model to predict forward rate of spread (km h$^1$) and flame height (m) for each site (masticated and control) under varying weather conditions. For ease of analysis, fuel moisture content (FMC) was held constant at 7% to represent severe fire conditions. Rate of spread and flame height were then predicted for zero wind and 30 km h$^{-1}$ wind conditions to represent contrasting wildfire scenarios. The significance of differences in the fire predictions between masticated and control plots were tested for normality and analysed using paired *t*-tests.

## 3. Results

### 3.1. Fuel Properties

Masticated plots had a lower density of taller shrubs (between 50–200 cm) (Figure 2). Non-overlapping confidence intervals in these height ranges and the *p*-values for the GAMs (Appendix A, Table A1) illustrate the substantial difference in shrub density. In contrast, there was little difference between 0–2 years and 3–4 years post mastication for shrub density above 50 cm. There was a significant reduction in mean vegetation closure following mastication (Figure 3). The difference was most pronounced within 0–2 years post mastication ($p < 0.01$; Appendix A, Table A2).

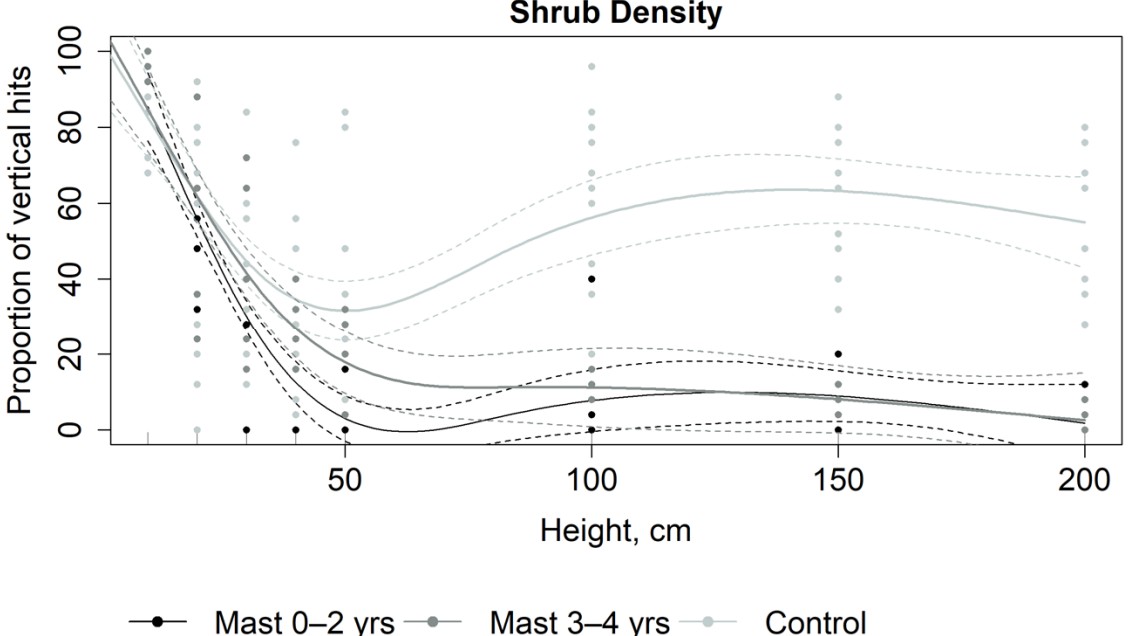

**Figure 2.** Generalised additive models showing mean proportion of vertical hits of fine fuel as a function of height (cm) for 0–2 years post mastication (black), 3–4 years post mastication (dark grey) and control (light grey). Dashed lines represent the 95% confidence interval around the mean. GAM diagnostics are provided in the Appendix A, Table A1.

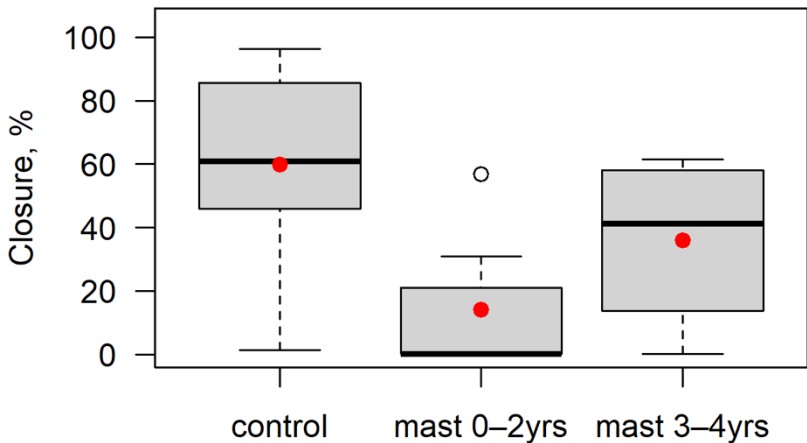

**Figure 3.** Boxplots of vegetation closure (%) measured from a height of 1.5 m for 0–2 years post mastication, 3–4 years post mastication and control. The mean is indicated by the red dot.

Fine and coarse dead surface fuel loads were significantly higher for 0–2 years post mastication than the control sites ($p = 0.001$ and $p < 0.001$, respectively; Appendix A, Table A2) (Figure 4a,b). Fine and coarse dead surface fuel loads were reduced for 3–4 years

post mastication, yet coarse fuels remained significantly higher than the control ($p < 0.05$, Appendix A, Table A2). Depth of surface fuel was higher in the masticated fuel beds and decreased over time; however, no significant differences were recorded (Appendix A, Table A2) (Figure 4c).

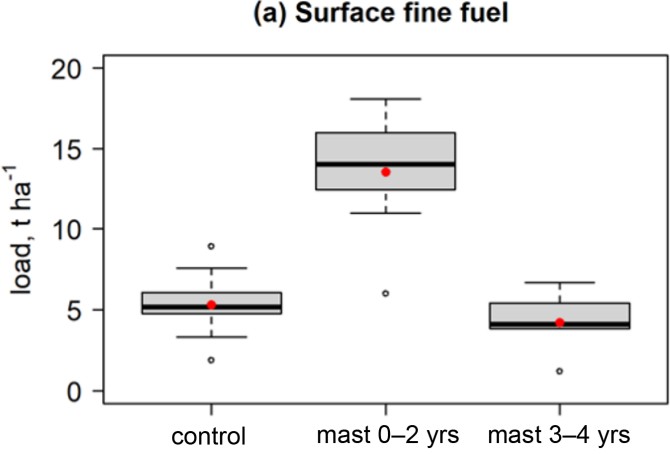

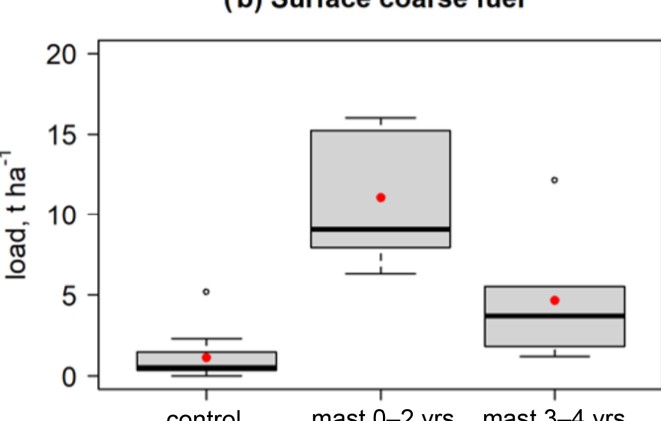

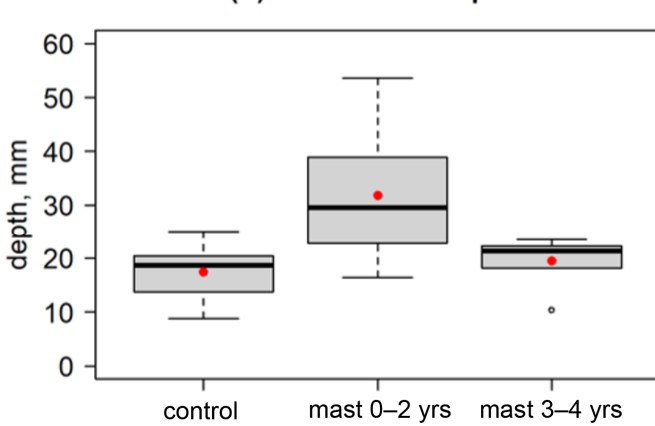

**Figure 4.** Box plots of dead surface fuel load (t haP$^{-1}$P) for (**a**) dead fine fuels, (**b**) dead coarse fuels and (**c**) surface fuel depth (mm) for 0–2 years post mastication, 3–4 years post mastication and control. The mean value is depicted by the red dot.

Differences in surface fuel loads are reflected in the contingency table (Figure 5), where the highest relative frequency of an 'extreme' surface fuel hazard score was significantly associated with the more recently masticated fuel beds ($p = 0.001$). Despite the increase in surface fuel hazard, the overall fuel hazard classes were lower in the masticated fuels compared with the control ($p = 0.001$), primarily due to the significant reduction in elevated fuel hazard.

## (a) Overall fuel hazard; Fisher's test *P* < 0.001

|  | Low | Moderate | High | Very High | Extreme |
|---|---|---|---|---|---|
| control | 0 | 0 | 0 | 5 | 8 |
| mast 0–2yrs | 0 | 2 | 4 | 0 | 1 |
| mast 3–4yrs | 0 | 3 | 3 | 0 | 0 |

## (b) Surface fuel hazard; Fisher's test *P* = 0.001

|  | Low | Moderate | High | Very High | Extreme |
|---|---|---|---|---|---|
| control | 0 | 10 | 3 | 0 | 0 |
| mast 0–2yrs | 0 | 2 | 0 | 1 | 4 |
| mast 3–4yrs | 2 | 3 | 0 | 1 | 0 |

## (c) Near surface fuel hazard; Fisher's test *P* = 0.171

|  | Low | Moderate | High | Very High | Extreme |
|---|---|---|---|---|---|
| control | 0 | 1 | 4 | 7 | 1 |
| mast 0–2yrs | 2 | 2 | 1 | 2 | 0 |
| mast 3–4yrs | 0 | 1 | 4 | 1 | 0 |

## (d) Elevated fuel hazard; Fisher's test *P* < 0.001

|  | Low | Moderate | High | Very High | Extreme |
|---|---|---|---|---|---|
| control | 0 | 0 | 0 | 6 | 7 |
| mast 0–2yrs | 3 | 4 | 0 | 0 | 0 |
| mast 3–4yrs | 1 | 5 | 0 | 0 | 0 |

## (e) Bark fuel hazard; Fisher's test *P* = 0.248

|  | Low | Moderate | High | Very High | Extreme |
|---|---|---|---|---|---|
| control | 3 | 7 | 2 | 0 | 1 |
| mast 0–2yrs | 5 | 1 | 0 | 0 | 1 |
| mast 3–4yrs | 2 | 4 | 0 | 0 | 0 |

**Figure 5.** Contingency tables showing the number of times each fuel hazard category was recorded in each treatment for (**a**) Overall fuel hazard; (**b**) Surface fuel hazard; (**c**) Near-surface fuel hazard; (**d**) Elevated fuel hazard; and (**e**) Bark fuel hazard. Pie charts provide a visual indicator of the relative frequency of occurrence of each hazard class within a treatment class.

### 3.2. Wildfire Behaviour

Predicted flame heights were significantly reduced in masticated fuels compared with the control for both the zero wind ($p < 0.01$ (0–2 years) and $p = 0.01$ (3–4 years); Appendix A, Table A2) and 30 km h$^{-1}$ wind speed ($p < 0.01$ (0–2 years) and $p < 0.05$ (3–4 years); Appendix A, Table A2) (Figure 6b,d). The extent of flame height reduction was more pronounced with greater windspeed. There were no significant differences predicted in the forward rate of spread for both wind speeds across the three treatments (Figure 6a,c).

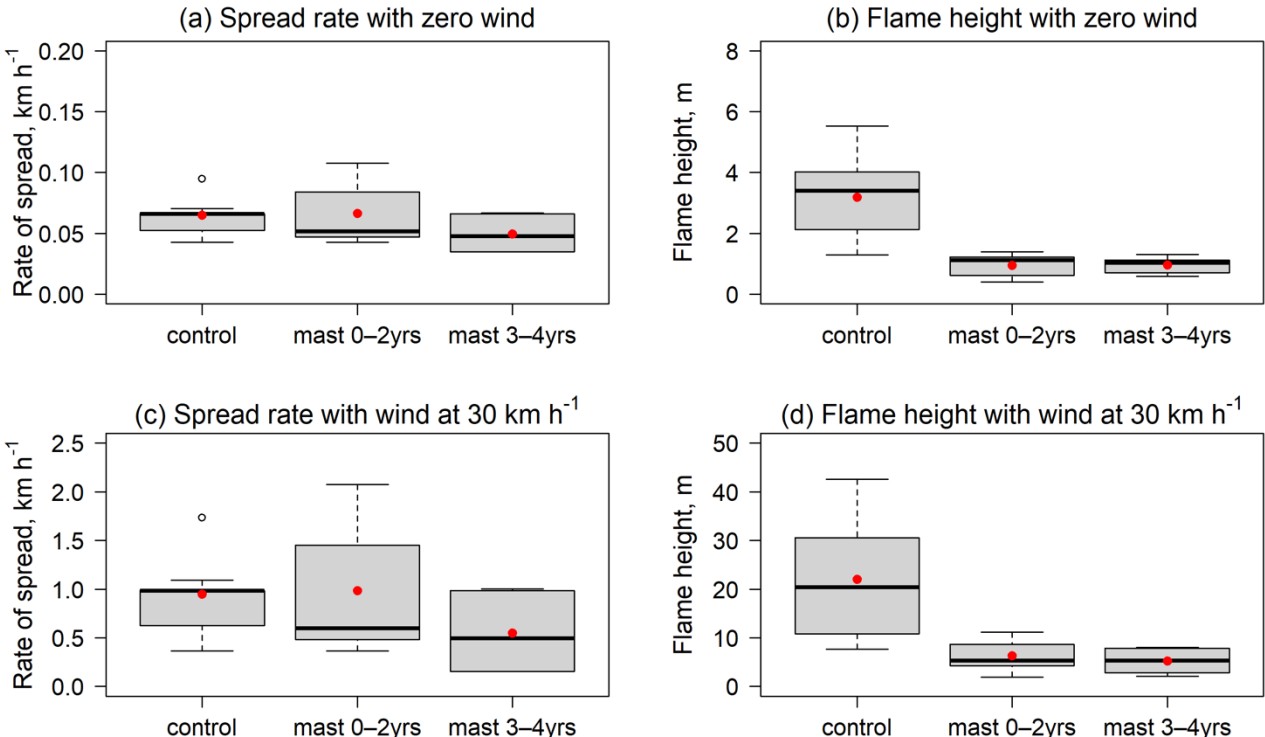

**Figure 6.** Box plots showing predicted forward rate of spread (km h$^{-1}$) for (**a**) zero wind and (**c**) 30 km h$^{-1}$ wind speeds and flame height (m) for (**b**) zero wind and (**d**) 30 km h$^{-1}$ wind speeds for 0–2 years post mastication, 3–4 years post mastication and control. The means are indicated by a red dot.

### 3.3. Floristic Diversity

Species richness and Shannon's diversity index were highest in plots 3–4 years post mastication; however, compared with the controls, these differences were not statistically significant (Appendix A, Table A2) (Figure 7). After excluding exotic species, plots 0–2 years post mastication showed the most notable reduction in both species' richness and Shannon's diversity index across the three treatments. Although, once again, compared with the controls, the difference was not statistically significant (Appendix A, Table A2).

Exotic species consisted predominately of graminoids and herbs. A significantly higher cover abundance of exotic herbs was recorded in the recently masticated plots compared with the controls ($p < 0.05$; Appendix A, Table A2) (Figure 8b). No significant differences were recorded for cover abundance of exotic graminoids across the three treatments. Most notable among the graminoids were *Ehrharta* sp. (present in eight plots, maximum cover 50%) and *Vulpia* sp. (present in 11 plots, maximum cover 50%). Most notable among the herbs were *Lysimachia arvensis* (present in 16 plots, maximum cover of 30%) and *Hypochoeris* sp. (present in 5 plots, maximum cover of 20%). The only exotic shrub species was *Chrysanthemoides monilifera*, which occasionally occurred in low abundance (e.g., 1–2% cover) in both the masticated and control plots.

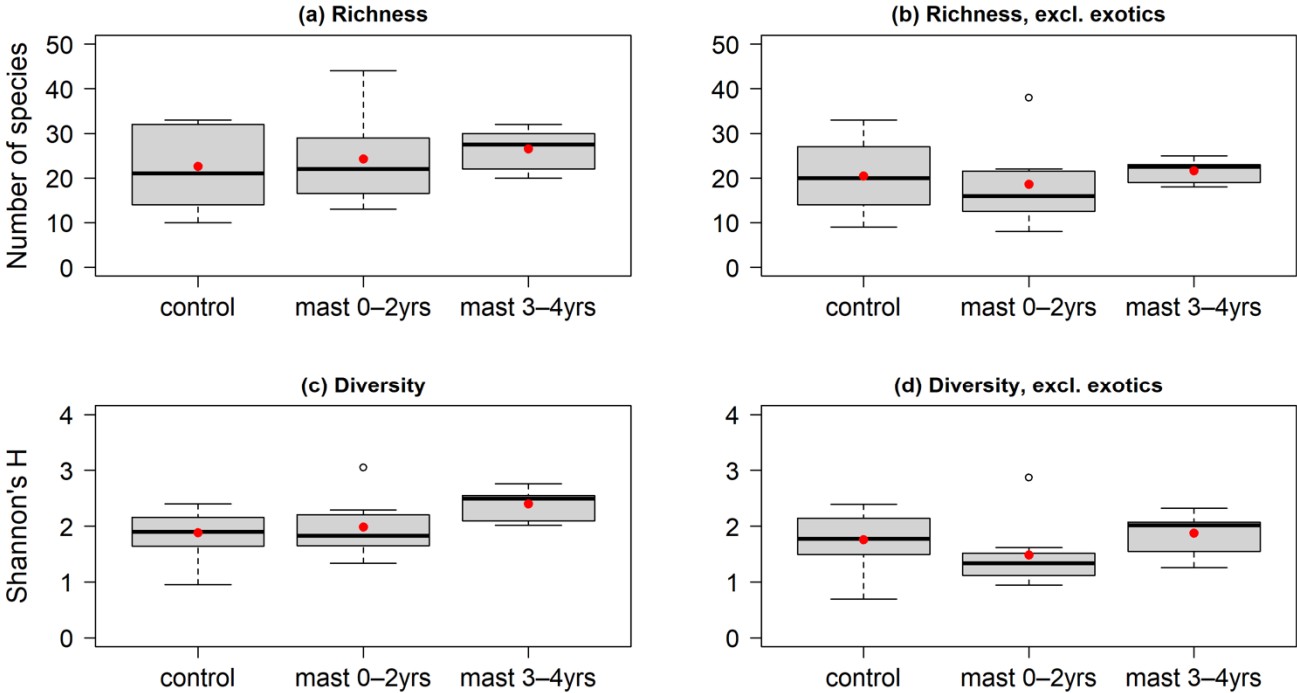

**Figure 7.** Floristic diversity metrics for (**a**) species richness; (**b**) species richness excluding exotic species; (**c**) Shannon's *H'*
and; (**d**) Shannon's *H'* excluding exotic species for 0–2 years since mastication, 3–4 years since mastication and control.

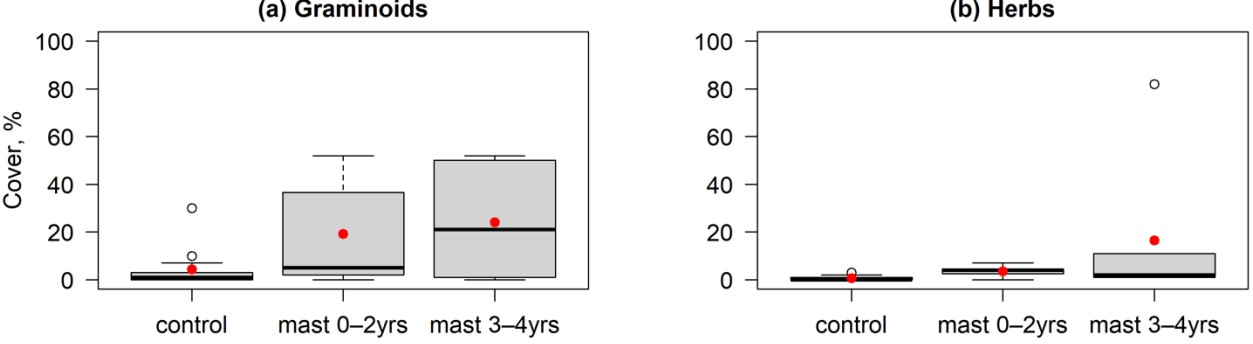

**Figure 8.** Cover abundance of exotic species: (**a**) graminoids; and (**b**) herbs for 0–2 years post mastication, 3–4 years post
mastication and control.

## 4. Discussion

Mastication in shrub encroached fuel significantly reduced shrub densities and potential flame heights for at least four years. This change in fire behaviour is likely to assist fire suppression efforts. Mastication caused little change to native species diversity but did introduce some exotic species.

### 4.1. Fuel Load and Structure

Mastication resulted in a deep layer of fuel on the forest floor. Unlike the naturally occurring fuel beds of the control sites, the coarse fraction in the masticated fuel bed was much higher and was still evident 3–4 years post treatment. Our combined average surface fuel loads were comparable to fuel loads reported for masticated sites in Californian chaparral [57] and in the lower ranges of previous studies in North American conifer forests [24,25,58]. This similarity in fuel loads with the chaparral is unsurprising as our sites more closely resemble the pre-masticated vegetation structure of chaparral shrublands than conifer forests.

Mastication produced a structural change to the vegetation community. The reduction in height and cover of woody understorey vegetation reduced the continuity of the fuel strata for at least four years. This is comparable to mastication in California chaparral where a significant decrease in woody shrub cover, compared with pre-treatment levels, was evident for three years or more [57,59]. In this study, a follow-up herbicide was applied to five out of the six older sites (3–4 years post mastication), which is likely to have influenced our findings and the longevity of the observed structural change (four years). In the same number of years, masticated-only gorse (*Ulex* spp.) shrublands exhibited a return to pre-treatment shrub cover [60]. The frequency of follow-up treatments (herbicide, mastication or prescribed fire) will be determined by site-specific factors including vegetation age [61], the re-sprouting ability of individual species [59] and the level of resources or funding available. Timing of mastication may also affect vegetation recovery, as highlighted by Potts et al. [59], where differences in seedling densities were observed between autumn and spring treatments. Further research is needed to quantify the individual effects of herbicide treatment and mastication to better understand the longevity of treatment effect and the environmental and human factors that influence these trends.

### 4.2. Wildfire Behaviour

Fuel continuity is important to fire behaviour. By reducing the horizontal and vertical continuity of the fuel, mastication limits the spread of flames between fuel elements and into the canopy [62,63], thereby reducing flame heights and fire severity. It is therefore unsurprising that the Vesta fire model predicted a reduction in flame heights for the masticated plots (compared with the control) as the shrub layer which links the surface fuels to the canopy had been removed. Lower flame heights aid fire suppression by reducing radiation exposure to firefighters, which makes direct attack and defence of infrastructure and assets safer and more achievable. At landscape scales, masticated fuel breaks provide opportunities for wildfire suppression by improving accessibility for direct attack [64] or enabling safer back burning operations [29]. Lower burn severities in masticated fuel contributes to the survivability of the vegetation, reducing both wind and water erosion, leading to faster recovery rates [28,29].

Fuel structure and load are significant factors in predicting fire behaviour. Hazard ratings for the surface (litter bed) and near surface (vegetation connected to the surface) fuel layers influence the rates of spread in the Dry Eucalypt Forest Model [20]. Surprisingly, we found no significant difference in rates of spread between the masticated and control plots, despite the significantly higher surface fuel hazards following mastication. It is possible that the higher levels of near surface fuel in the control plots compensated for their lower surface fuel hazard, resulting in similar rates of spread for both treatments overall. Near surface fuel has been shown to strongly influence spread rates in some eucalypt forests [65,66].

Empirical fire models rarely capture the complex interactions of factors that influence fire behaviour. In the context of mastication, a key limitation of existing Australian fire spread models is that they do not consider the influence of coarse fuel [55]. Coarse fuel is likely to influence aspects of fire behaviour including spread rate, fuel consumption, flaming and smouldering durations and ignition likelihood (as reviewed by [23]). Longer flaming and smouldering times may also have implications for increased soil heating [67] and the mortality rate of re-spouters. Reduction in vegetation closure is another feature of masticated sites not captured by existing models. This could be important for in-forest wind speeds [68] and fuel moisture contents [69], both of which are important determinants of fire rate of spread.

### 4.3. Species Richness and Diversity

Physical changes to vegetation strata are likely to have an impact on overall vegetation composition. The most prominent ecological change to the flora following mastication was the reduction in height and cover of the encroaching shrub species—an effect that was

likely prolonged with targeted herbicide application. An increase in species diversity was expected due to the reduction in competition for light, water and nutrients [70], resulting from the removal of the dominant shrub layer. However, mastication appeared to have no effect on the overall richness and diversity of native flora species. Previous research has reported similar findings of little to no effect of mastication on understorey plant richness in conifer forests [71] and shrub species richness and cover in chaparral [59]. On the other hand, more recent studies have observed an increase in species richness and cover, particularly herbaceous species, both native and exotic [72,73]. Monitoring over a longer time frame is needed to make these conclusions with greater certainty as vegetation growth may be hampered for several years by the dense layer of masticated material on the forest floor [74]. Longer periods of shrub encroachment may also deplete the richness and composition of the germinable soil seedbank [13]. Furthermore, many species in these communities are fire responsive, needing heat, smoke or both to germinate. Mechanical fuel treatments provide neither of these, so investigations into follow-up burn treatments or the application of smoke compounds may be warranted.

Mastication created an environment which is suitable for the establishment of exotic species. There was a significant increase in the cover abundance of exotic herbs in the initial years following mastication. Although herbicide application was intended for the encroaching shrub, this treatment may have had an impact on the presence and abundance of exotic species. Previous studies have also highlighted the tendency for exotic species to establish post mastication [43,71,72,75]. Monitoring over a longer timeframe is needed to determine whether this is a lasting effect. The prevalence of exotic species may affect native flora success in the longer term [75], or conversely, native flora may eventually outcompete short-lived exotic species. Fornwalt et al. [73] emphasise that observations of the positive trend in native understory plant cover and diversity were most pronounced 6–9 years post mastication.

In addition to the floristic implications of mastication, there could also be faunal impacts that warrant consideration. Some faunal species favour dense cover created by shrub encroachment (e.g., wallabies) [15] with a greater structural complexity for perching or nesting [16]. Others prefer a more open habitat with less physical restriction (e.g., kangaroos) [76] or an increased abundance of groundcover (e.g., granivorous birds) [77], which may increase in abundance following mastication. In other studies, populations of pest species have been found to benefit initially after mechanical treatment, the European rabbit [76] and the house mouse [78]. Mastication treatments will require an approach that considers the possible impact on faunal populations. Possible management strategies include creating refuge areas or allowing vegetation to become more structurally complex in between treatments [78].

### 4.4. Implications for Fire Management

The potential for mastication to reduce flame heights is likely to increase opportunities for wildfire mitigation and response. However, follow-up herbicide treatment may be required for the ongoing management of shrub-encroached ecosystems and their elevated fire risk. Mastication provides an alternative to prescribed burning, especially for fuel types that are difficult to burn, either due to their structure or their location on the wildland-urban interface (e.g., dense shrub fuels with are unlikely to burn under low wind prescribed burning conditions). The window of opportunity to implement safe and effective prescribed burns is changing [79] as fire seasons lengthen due to warmer temperatures and drought under climate change [80]. Mechanical mastication can be applied almost year-round and may become increasingly important.

The cost and labour intensiveness of mastication is generally higher than prescribed burning and even greater if it requires follow-up herbicide. Therefore, it is crucial that treatment locations are optimised [32,81]. Fewer strategically located fuel treatments may be more effective than a greater overall area of treated fuel [64]. For land managers to evaluate treatment options, they need to have realistic expectations about the reductions in

wildfire risk that are likely to result from mastication. This is important in conjunction with other treatments such as prescribed burning and suppression, which function on different spatial and temporal scales [82].

## 5. Conclusions

Mastication changed fuel structure and predicted fire behaviour in eucalypt woodlands and forests, consistent with comparatively similar research worldwide. Further work is required to understand how the masticated surface fuels will influence wildfire behaviour, the longevity of treatment effect and the need for follow up. Application in key areas is likely to reduce risk to some assets, although the cost trade-off requires exploration. Mastication increases exotics in the short-term but long-term studies are required to understand the changes to community composition, as well as the return of the invasive shrubs. The potential impacts of mastication, herbicides and the combination of the two will require further examination. It is imperative that land managers are provided with this knowledge to make informed decisions about management and resource allocation to ensure the safety of communities and the endurance of natural systems.

**Author Contributions:** M.A.G. and J.G.C. conceived and designed the study with contributions from T.J.D., T.D.P. and B.J.P.; M.A.G. and B.J.P. performed the field data collection; M.A.G., J.G.C., T.J.D. and T.D.P. analysed the data; M.A.G. prepared the original draft; the review and editing process involved J.G.C., T.J.D., T.D.P. and B.J.P. All authors have read and agreed to the published version of the manuscript.

**Funding:** This research was undertaken as an Honours research project with funding from the Department of Environment, Land, Water and Planning.

**Data Availability Statement:** All data are stored on a central repository at the University of Melbourne and are available from the corresponding author on reasonable request.

**Acknowledgments:** Thank you to Dougal McAllister for his assistance with field data collection. Field samples were collected under Parks Victoria permit number 10008751. We also acknowledge the fire officers and analysts working for both DELWP and Parks Victoria who shared their knowledge and experience.

**Conflicts of Interest:** The authors declare no conflict of interest.

## Appendix A

**Table A1.** Diagnostics of the GAM models developed for shrub density for masticated and control. Statistical significance denoted by *** $p \leq 0.001$.

|  | df | F | *p*-Value | RP$^2$P (Adjusted) | Deviance Explained |
|---|---|---|---|---|---|
| control | 2.97 | 316.06 | <0.001 *** | 0.31 | 33% |
| mast 0–2 years | 2.98 | 349.97 | <0.001 *** | 0.79 | 74% |
| mast 3–4 years | 2.97 | 361.48 | <0.001 *** | 0.80 | 81% |

**Table A2.** Results of paired *t*-test or nonparametric Wilcoxon signed-rank test. Includes degrees of freedom (*df*), test statistic (*t*) and *p*-value. Statistical significance denoted by * $p \leq 0.05$; ** $p \leq 0.01$; *** $p \leq 0.001$.

|  | Mast 0–2 yrs—Control | | | Mast 3–4 yrs—Control | | |
|---|---|---|---|---|---|---|
|  | *df* | *t* | *p*-Value | *df* | *t* | *p*-Value |
| Vegetation closure | 6 | −6.245 | 0.001 *** | 5 | −2.166 | 0.083 |
| Surface dead fine fuel load | 6 | 5.532 | 0.001 *** | 5 | −0.533 | 0.616 |
| Surface dead coarse fuel load | 6 | 6.871 | <0.001 *** | 5 | 21R $_W$ | 0.031 * |
| Surface fuel depth | 6 | 2.304 | 0.061 | 5 | 1.378 | 0.227 |
| Spread rate @ 0 km h$^{-1}$ wind | 6 | −0.394 | 0.707 | 5 | −1.298 | 0.251 |

**Table A2.** *Cont.*

| | Mast 0–2 yrs—Control | | | Mast 3–4 yrs—Control | | |
|---|---|---|---|---|---|---|
| | *df* | *t* | *p*-Value | *df* | *t* | *p*-Value |
| Flame ht @ 0 km h$^{-1}$ wind | 6 | −4.980 | 0.003 ** | 5 | −4.079 | 0.010 ** |
| Spread rate @ 30 km h$^{-1}$ wind | 6 | −0.394 | 0.707 | 5 | −1.298 | 0.251 |
| Flame height @ 30 km h$^{-1}$ wind | 6 | −4.368 | 0.005 ** | 5 | −3.707 | 0.014 * |
| Species richness | 6 | −0.131 | 0.900 | 5 | 1.845 | 0.124 |
| Species richness excl. exotics | 6 | −1.271 | 0.251 | 5 | 1.047 | 0.343 |
| Shannon's H | 6 | 0.647 | 0.542 | 5 | 3.653 | 0.0147 |
| Shannon's H excl. exotics | 6 | −1.127 | 0.303 | 5 | 0.454 | 0.669 |
| Cover exotic graminoids | – | 15R $_W$ | 0.059 | – | 15R $_W$ | 0.059 |
| Cover exotic herbs | 6 | 3.447 | 0.014 * | – | 15R $_W$ | 0.059 |

w Indicates that a Wilcoxon sign test was used instead of the *t*-test and for those instances; V—the sum of ranks associated with positive differences, is reported rather than the *t*-statistic.

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
