# Peer review of "Mechanical Mastication Reduces Fuel Structure and Modelled Fire Behaviour in Australian Shrub Encroached Ecosystems"

_forests, doi:10.3390/f12060812_

Round 1

Reviewer 1 Report

Thank you for this submission to Forests. I believe this study is an appropriate fit for this journal and the content will be appealing to a wide array of readers. I feel you have placed great time and effort into this work and you should be commended for that. I did not find major grammatical or structural issues presented by your writing - only a few missing commas and choices that need to be made regarding the inclusion or exclusion of hyphens with some of the phrases used throughout the manuscript. 

I do have a rather major concern that needs to be addressed: not all of the treatments were mastication only. Some were mixed with herbicide treatments. Most appropriately, those treatments should be accounted for separately in the statistical analyses. At this point, you cannot accurately assess whether mastication created the measured effects you describe if, in fact, some of the treatments were not mastication-only. If you cannot account for those as a separate treatment, perhaps stating in the Discussion that the results might be influenced by the mixed-treatment would be appropriate? How did the herbicides influence specific responses? Perhaps unique features/attributes noted in units that were treated with this combination might be described in a paragraph (or two) in the Discussion? I think you must elaborate on this point before the manuscript can move forward to publication. 

I believe this will constitute a minor revision and resubmission. I have attached my review of the entire manuscript as a separate PDF.

Author Response

Response to reviewers' comments

 Reviewer 1:

Thank you for this submission to Forests. I believe this study is an appropriate fit for this journal and the content will be appealing to a wide array of readers. I feel you have placed great time and effort into this work and you should be commended for that. I did not find major grammatical or structural issues presented by your writing - only a few missing commas and choices that need to be made regarding the inclusion or exclusion of hyphens with some of the phrases used throughout the manuscript. 

>> We have changed minor typographical issues identified and outlined in the accompanied pdf. This includes the choice to consistently exclude hyphens for the term “shrub encroachment”.

I do have a rather major concern that needs to be addressed: not all of the treatments were mastication only. Some were mixed with herbicide treatments. Most appropriately, those treatments should be accounted for separately in the statistical analyses. At this point, you cannot accurately assess whether mastication created the measured effects you describe if, in fact, some of the treatments were not mastication-only. If you cannot account for those as a separate treatment, perhaps stating in the Discussion that the results might be influenced by the mixed-treatment would be appropriate? How did the herbicides influence specific responses? Perhaps unique features/attributes noted in units that were treated with this combination might be described in a paragraph (or two) in the Discussion? I think you must elaborate on this point before the manuscript can move forward to publication. 

>> Since mastication is a relatively new technique in Victoria, there were not enough sites available (insufficient replication) to independently evaluate the effect of herbicide treatment. As such we considered the follow-up herbicide a part of the mastication treatment. This is a limitation to the study.

We do agree that herbicide could be a contributing factor in reducing the density of taller shrubs.

We have made the following changes to the report:

Table 1 has been rearranged in order of treatment year to effectively highlight the mastication sites that were subject to follow up herbicide – five out of the six older sites (3-4 years post mastication).

Line 157: “As we were unable to quantify the individual effects of mastication-only versus mastication plus herbicide, we considered the follow up herbicide a part of the mastication treatment.”

We have reworded the discussion to state (line 400):

“Mastication produced a structural change to the vegetation community. By reducing the height and cover of woody understorey vegetation, it reduced the vertical and horizontal continuity of the fuel strata for at least four years. This is comparable to mastication in California chaparral where a significant decrease in woody shrub cover, compared with pre-treatment levels, was evident for 3 years or more [57, 59]. However, follow up herbicide application was considered a component of the mastication treatment in this study and this is likely to have influenced our findings and the longevity of the observed structural change, 4 years. In the same number of years, masticated gorse (Ulex spp) shrublands exhibit a return to pre-treatment shrub cover [60]. The frequency of follow up treatments (herbicide, mastication or prescribed fire) will be determined by site specific factors including vegetation age [61] and re-sprouting ability of individual species [59] as well as the level of resources or funding available. Timing of mastication may also affect vegetation recovery, as highlighted by Potts et al. [59] where differences in seedling densities were observed between autumn and spring treatments. Further research is needed to quantify the individual effects of herbicide treatment and mastication to better understand the longevity of treatment effect and the environmental and human factors that influence these trends.”

I believe this will constitute a minor revision and resubmission. I have attached my review of the entire manuscript as a separate PDF.

Reviewer 2 Report

Overall, the authors have presented interesting results from a study about a topic that is relevant fire management. The content of the manuscript is pertinent for both the scientific community and the fire managers.

Some suggestions and comments to improve the current content of the manuscript are as follows:

Line 21: What was about mastication without herbicide treatment? Could it also contribute to reduce the density of taller shrubs? Only five of the thirteen sites were treated by herbicide.

Line 23-24: “… increase in fine fuel loads”. I think that it should be increase in fine dead fuel loads.

Line 82. It is an unpublished document. Why was more easily adjacent to masticated fuel? Is it related to flame length or fire-line construction?

 Table 1. What is the season of each treatment? Could the treatment season influence in the species growth or regeneration level?

Line 163, line 170 and line 181. What was the fine fuel size? 2 cm, 2.5 cm, or 3 cm?

Line 182. I think that you used 25 measurements within each plot to identify the presence of fine fuel. In this sense, why do you use only 12 measurements?

Table 2. What are 1-hour and 10-hour classes? These terms have not been previously defined.

Table 3. I think that percentage could be more interesting than the number of times each fuel hazard category.

3.2. Wildfire behaviour. I think that it is important to point out that flame length differences are more marked by 30 km h-1 scenario. In the 0 km h-1 scenario, mast 0-2 yrs and mast 3-4 yrs allowed us to directly attack the flames (< 2.5 m).

Line 414. What are fuel models that are difficult to burn?

Author Response

Response to reviewers' comments

Reviewer 2:

Overall, the authors have presented interesting results from a study about a topic that is relevant fire management. The content of the manuscript is pertinent for both the scientific community and the fire managers.

Some suggestions and comments to improve the current content of the manuscript are as follows:

Line 21: What was about mastication without herbicide treatment? Could it also contribute to reduce the density of taller shrubs? Only five of the thirteen sites were treated by herbicide.

>> Refer to our responses to reviewer 1 in relation to herbicide treatment (provided below).

Since mastication is a relatively new technique in Victoria, there were not enough sites available (insufficient replication) to independently evaluate the effect of herbicide treatment. As such we considered the follow-up herbicide a part of the mastication treatment. This is a limitation to the study.

We do agree that herbicide could be a contributing factor in reducing the density of taller shrubs.

We have made the following changes to the report:

Table 1 has been rearranged in order of treatment year to effectively highlight the mastication sites that were subject to follow up herbicide – five out of the six older sites (3-4 years post mastication).

Line 157: “As we were unable to quantify the individual effects of mastication-only versus mastication plus herbicide, we considered the follow up herbicide a part of the mastication treatment.”

We have reworded the discussion to state (line 400):

“Mastication produced a structural change to the vegetation community. By reducing the height and cover of woody understorey vegetation, it reduced the vertical and horizontal continuity of the fuel strata for at least four years. This is comparable to mastication in California chaparral where a significant decrease in woody shrub cover, compared with pre-treatment levels, was evident for 3 years or more [57, 59]. However, follow up herbicide application was considered a component of the mastication treatment in this study and this is likely to have influenced our findings and the longevity of the observed structural change, 4 years. In the same number of years, masticated gorse (Ulex spp) shrublands exhibit a return to pre-treatment shrub cover [60]. The frequency of follow up treatments (herbicide, mastication or prescribed fire) will be determined by site specific factors including vegetation age [61] and re-sprouting ability of individual species [59] as well as the level of resources or funding available. Timing of mastication may also affect vegetation recovery, as highlighted by Potts et al. [59] where differences in seedling densities were observed between autumn and spring treatments. Further research is needed to quantify the individual effects of herbicide treatment and mastication to better understand the longevity of treatment effect and the environmental and human factors that influence these trends.”

Line 23-24: “… increase in fine fuel loads”. I think that it should be increase in fine dead fuel loads.

>> This is correct. The inclusion of “dead” to fine fuels occurred throughout the document where appropriate.

Line 82. It is an unpublished document. Why was more easily adjacent to masticated fuel? Is it related to flame length or fire-line construction?

>> We have altered the sentences here for clarity. They now read (at line 84):

“Direct attack was not possible in the masticated or untreated shrubby areas, but crews were able to defend houses more easily adjacent to masticated fuel because lower flame heights made asset protection safer to implement [35].”

Table 1. What is the season of each treatment? Could the treatment season influence in the species growth or regeneration level?

>> Since mastication is a relatively new technique in Victoria, there were not enough sites or data available to independently evaluate the effect of treatment season. Line 154 has been updated to include this point.

I do agree that treatment season may influence species growth or regeneration level. This is highlighted in the discussion, line 411.

Line 163, line 170 and line 181. What was the fine fuel size? 2 cm, 2.5 cm, or 3 cm?

>> This section has been altered to make it easier to understand. Live fine fuel is < 2mm thick (line 191), dead fine fuel is < 6 mm thick (line 192, 201), dead coarse fuel is ≥ 6 diameter < 25 mm (line 202, 199). Repeated definitions were removed from line 213.

Line 182. I think that you used 25 measurements within each plot to identify the presence of fine fuel. In this sense, why do you use only 12 measurements?

>> This section has been edited for clarity. Line 193:

“The presence of fine fuel at each height range was summarised as a proportion of the total 25 measurements taken within each plot.”

And line 217: “Such as, surface fuel depth which was measured 12 times within the plot...”

Table 2. What are 1-hour and 10-hour classes? These terms have not been previously defined.

>> This has been changed to Dead fine (< 6 mm diameter) / Dead course (6 ≥ diameter < 25 mm) which is consistent with the definitions outlined at lines 192, 199, 201 and 202.

Table 3. I think that percentage could be more interesting than the number of times each fuel hazard category.

>> This is a fair point. We explored this recommendation; however, we found the end result was less visually appealing/intuitive than our original infographic.

3.2. Wildfire behaviour. I think that it is important to point out that flame length differences are more marked by 30 km h-1 scenario. In the 0 km h-1 scenario, mast 0-2 yrs and mast 3-4 yrs allowed us to directly attack the flames (< 2.5 m).

>> We have added this point, line 344: “The extent of flame height reduction was more pronounced with greater windspeed.”

Line 414. What are fuel models that are difficult to burn? Don’t

>> An example has been included. The sentence now reads at line 499:

“Mastication provides an alternative to prescribed burning, especially for fuel types that are difficult to burn, either due to their structure or their location on the wildland-urban interface (e.g. dense shrub fuels are unlikely to burn in low wind prescribed burning conditions).”

Round 2

Reviewer 1 Report

Thank you for your revisions. I still feel that additional clarification is needed regarding the impacts of mastication in the Introduction, section 4.4, and Conclusion. Additional research will be needed to define the potential impacts of mastication, herbicides, and the combination of the two. As you stated in your revision report, there weren’t enough experimental units to determine 1) how herbicides are creating their own effects or 2) adding to the effects of mastication. I do not believe this is a fatal flaw of the research, but you do need to specify the potential impacts throughout the manuscript, not just in one section. Once these changes are made, the manuscript should be considered for acceptance. Thank you again for your work to improve this manuscript. 

Author Response

Response to reviewer 1’s comments(round 2):

Thank you for your revisions. I still feel that additional clarification is needed regarding the impacts of mastication in the Introduction, section 4.4, and Conclusion. Additional research will be needed to define the potential impacts of mastication, herbicides, and the combination of the two. As you stated in your revision report, there weren’t enough experimental units to determine 1) how herbicides are creating their own effects or 2) adding to the effects of mastication. I do not believe this is a fatal flaw of the research, but you do need to specify the potential impacts throughout the manuscript, not just in one section. Once these changes are made, the manuscript should be considered for acceptance. Thank you again for your work to improve this manuscript. 

>> Thank you for your valuable feedback. We have included a number of references to the inclusion of follow up herbicide and provided comments on the possible implications. 

Please note, tracked changes were posing significant formatting issues. As such, all previous tracked changes have been accepted and the most recent edits for this revision are highlighted in yellow. If this is an unsuitable format please have Ms Wu contact me - thank you.

We have made the following changes to the manuscript:

Line 89: “ Follow up herbicide was considered a component of the mastication treatment where the encroaching shrub species were re-establishing.”

Line 352: “In this study, follow up herbicide was applied to five out of the six older sites (3-4 years post mastication) which is likely to have influenced our findings and the longevity of the observed structural change (four years).”

Line 400: “… the reduction in height and cover of the encroaching shrub species – an effect that was likely prolonged with targeted herbicide application.”

Line 419: “Although herbicide application was intended for the encroaching shrub, this treatment may have had an impact on the presence and abundance of exotic species.”

Line 441: “However, follow up herbicide treatment may be required for the ongoing management of shrub encroached ecosystems and their elevated fire risk.”

Line 450: “The cost and labour intensiveness of mastication is generally higher than prescribed burning and even greater if it requires follow up herbicide.”

Line 467: “The potential impacts of mastication, herbicides and the combination of the two will require further examination.”
